**Subject Area:**
developmental biology/genetics/molecular biology/cellular biology

Cdkn1b/p27, Cdkn1c/p57, differentiation, Gata3, lens, Prox1

**Author for correspondence:**
Ales Cvekl
e-mail: ales.cvekl@einstein.yu.edu

# Transcriptomic analysis and novel insights into lens fibre cell differentiation regulated by Gata3

Elena Martynova[1], Yilin Zhao[1], Qing Xie[1], Deyou Zheng[2] and Ales Cvekl[1]

[1]Departments of Ophthalmology and Visual Sciences and Genetics, and [2]Departments of Genetics, Neurology, and Neuroscience, Albert Einstein College of Medicine, Bronx, NY 10461, USA

  AC, 0000-0002-3957-789X

Gata3 is a DNA-binding transcription factor involved in cellular differentiation in a variety of tissues including inner ear, hair follicle, kidney, mammary gland and T-cells. In a previous study in 2009, Maeda *et al.* (*Dev. Dyn.* **238**, 2280–2291; doi:10.1002/dvdy.22035) found that Gata3 mutants could be rescued from midgestational lethality by the expression of a Gata3 transgene in sympathoadrenal neuroendocrine cells. The rescued embryos clearly showed multiple defects in lens fibre cell differentiation. To determine whether these defects were truly due to the loss of Gata3 expression in the lens, we generated a lens-specific Gata3 loss-of-function model. Analogous to the previous findings, our Gata3 null embryos showed abnormal regulation of cell cycle exit during lens fibre cell differentiation, marked by reduction in the expression of the cyclin-dependent kinase inhibitors Cdkn1b/p27 and Cdkn1c/p57, and the retention of nuclei accompanied by downregulation of Dnase IIβ. Comparisons of transcriptomes between control and mutated lenses by RNA-Seq revealed dysregulation of lens-specific crystallin genes and intermediate filament protein Bfsp2. Both Cdkn1b/p27 and Cdkn1c/p57 loci are occupied *in vivo* by Gata3, as well as Prox1 and c-Jun, in lens chromatin. Collectively, our studies suggest that Gata3 regulates lens differentiation through the direct regulation of the Cdkn1b/p27 and Cdkn1c/p57 expression, and the direct/or indirect transcriptional control of Bfsp2 and Dnase IIβ.

## 1. Introduction

The regulation of cell proliferation and cell cycle exit-coupled differentiation is a central feature of embryonic development and organogenesis. The critical components of cell cycle regulation are highly conserved in metazoan organisms (see [1,2]). Disruption of appropriate regulation can result in growth defects, tissue malfunction and/or cancer.

The ocular lens has served as a leading experimental model to study cell cycle exit for over three decades (see [3,4]). Cell cycle exit during terminal differentiation helps to establish the polarity of the embryonic lens vesicle at E11.5. The posterior portion of the lens vesicle is exposed to bone morphogenic protein (BMP) and fibroblast growth factor (FGF) growth factors originating from the optic cup/prospective neuroretina, and these factors are known to induce cell cycle exit-coupled lens fibre cell differentiation (reviews: [5–7]). By contrast, the anterior portion of the vesicle differentiates into the lens epithelium, comprising quiescent and proliferative cells [8]. Genetic studies demonstrated that temporally and spatially upregulated expression of two negative regulators of cyclin-dependent kinases, Cdkn1b/p27 and Cdkn1c/p57, is critical for the proper initiation of lens fibre cell differentiation [9–11]. A hallmark feature of lens fibre cell maturation is the temporally and spatially controlled degradation of the intracellular organelles, the mitochondria, endoplasmic reticulum, Golgi apparatus and finally the nuclei [12–14]. Nuclear degradation involves the activities of a number of

DNA-binding transcription factors, DNA repair proteins Ddb1 and Nbs1, chromatin remodelling Brg1 and Snf2 h ATPases, and various enzymes, such as Alox15 and DNase IIβ [15,16]. Prior to their physical destruction, the nuclei change their shape from ovoid to more rounded, reduce their sizes and transfer both histone and non-histone proteins into the cytoplasm [17]. Retention of nuclei increases light scattering and triggers cataractogenesis [15].

Disruption of BMP and FGF signalling in lens, evaluated through loss-of-function experiments of genes encoding BMP and FGF receptors, negatively affects the expression of both Cdkn1b and Cdkn1c mRNAs and/or proteins [18–22]. Both FGFs [23,24] and BMPs [25] promote *in vitro* lens fibre cell differentiation. Subsequent *in vitro* studies also demonstrated that FGF and BMP signalling pathways in lens cells are interactive in that BMP keeps lens cells in an optimally FGF-responsive state, and, reciprocally, FGF enhances BMP-mediated gene expression [26–28]. Studies of multiple DNA-binding transcription factors in the lens, including Gata3 [29], Pax6 [30], Prox1 [31,32] and Pitx3 [33,34], revealed their roles in early stages of fibre cell differentiation and cell cycle exit control [7]. Other transcription factors, such as c-Maf, Hsf4 and Sox1, control crystallin gene expression [7], while compound loss of function of MafG and MafK serves as a model for age-onset cataractogenesis [35]. Downregulation of Cdkn1b and Cdkn1c, visualized by antibodies or *in situ* hybridizations, was found in *Gata3*, *Pitx3* and *Prox1* mutants [29–32,34]. Nevertheless, it is not known how these transcription factors respond to extracellular signalling in the lens and whether they are directly or indirectly involved in transcriptional control of *Cdkn1b* and *Cdkn1c* genes.

Analysis of the expression domains of a number of DNA-binding transcription factors regulating lens development identified restricted expression in the posterior part of the lens vesicle for c-Jun [36] and Gata3 [29,37]. By contrast, the majority of these factors, excluding Hsf4 and Sox1, which are turned on later in differentiating lens fibres [7], are expressed in both compartments of the lens vesicle. Gata3 expression is also detected in the surface ectoderm that gives rise to the lens placode formed by lens progenitor cells [29,36]. Prox1 is also highly upregulated in the nuclei of the posterior lens vesicle; however, it is also expressed in the anterior portion of the lens, both in the cytoplasm and nuclei [32,38]. FGF-regulated factors Etv1 (ER81) and Etv5 (ERM) and BMP-regulated Smads are also differentially expressed in the early lens vesicle and differentiating lens fibres [21,39–41].

Gata3 belongs to the GATA family of transcription factors that bind consensus 5′-(A/T)GATA(A/G)-3′ DNA sequences in the promoters and enhancers of target genes [42–44]. Gata3 plays a crucial role in the embryonic development. Gata3 null embryos die around E11.5 due to internal bleeding [45]. Transgene-mediated restoration of Gata3 expression in sympathoadrenal lineages using the human dopamine β-hydroxylase promoter was subsequently shown to be sufficient to rescue the embryonic lethality [29]. Using this genetic rescue approach, Maeda *et al.* [29] assessed lens development and discovered abnormal proliferation and failure to undergo proper cell cycle exit during fibre cell differentiation. The mutant embryos showed reduced expression of p27 and p57 at the mRNA levels. This experimental model, however, does not directly address the issue of whether Gata3 expression could be essential for the earliest stages of lens progenitor specification, or whether Gata3 is required only autonomously for proper fibre cell differentiation.

In this study, we report lens-specific conditional inactivation of *Gata3* using a Pax6-Cre mouse line coupled with global transcriptome analysis by RNA-Seq. Gata3-depleted lenses exhibit abnormal cell cycle exit, altered lens fibre cell morphology, denucleation defects and cataract formation. These abnormalities were visible by E12.5, and analysis of the expression changes of E14.5 mutant lenses revealed downregulation of mRNAs encoding specific β/γ-crystallins, DNase IIβ, phakinin/Bfsp2, Hopx and other genes. Expression of Cdkn1b/p27 and Cdkn1c/p57 proteins was also downregulated in *Gata3* null lenses, and direct Gata3 binding was observed at both *Cdkn1b/p27* and *Cdkn1c/p57* upstream promoter regions. Taken together, these data suggest that Gata3 regulates cell cycle exit coupled differentiation of lens fibre cells via a direct transcriptional regulation of Cdkn1b/p27 and Cdkn1c/p57 expression.

# 2. Material and methods

## 2.1. Conditional inactivation of Gata3 in lens progenitor cells

The conditional *Gata3* floxed allele (in the C57BL/6 background) was generated through homologous recombination as described elsewhere [46], and *Gata3^{f/f}* mice were kindly provided by Dr Jinfang Zhu from the National Institute of Allergy and Infectious Diseases, the National Institutes of Health, Bethesda, MD. The *Gata3* null allele was obtained by the deletion of exons 4 that introduces a reading frame shift that would prevent the expression of exon 5 and more distal exons. Mice carrying the *Gata3*-floxed, wild-type allele or the *Gata3*-deleted allele were identified by polymerase chain reaction (PCR) with the following primer pairs: P13 and P16; and P8 and P16: 5′-TCAGGGCACTAAGGGTTGTTAACTT-3′ (P8), 5′-GAATTCCATCCATGAGACACACAA-3′ (P11), 5′-CAGTCTCTGGTATTGATCTGCTTCTT-3′ (P13), 5′-GTGCAGCAGAGCAGGAAACTCTCAC-3′ (P16). To inactivate *Gata3* specifically in lens progenitor cells, we used the Pax6-Cre mouse line described previously [47]. The Pax6-Cre line differs from the similar earlier line, le-cre, in the regulatory sequences driving the Cre, absence of GFP and probably the genomic integration site [48]. Mice were screened by PCR using tail genomic DNA and Cre-specific primers, forward: 5′-ATGCTTCTGTCCGTTTGCC-3′ and reverse: 5′-CAACAC-CATTTTTTCTGACCC-3′, yielding a 650 bp product. Gata3 CKOs (*Gata3^{f/f}*, *Pax6Cre*/+) and their control littermates were generated by crossing the *Gata3^{f/f}* with the *Gata3^{f/wt}*, *Pax6Cre*/+ mice. Noon of the day, the vaginal plug was detected and was considered E0.5.

## 2.2. Haematoxylin and eosin staining and gross histology

Dissected embryos were fixed in 4% paraformaldehyde overnight and were then dehydrated in an ethanol gradient, processed and embedded in paraffin at the Histology and Comparative Pathology Core Facility at the Albert Einstein College of Medicine, New York. Transverse sections were produced at 5 µm using a microtome. The slides were incubated for 1 h at 60°C, deparaffinized in xylene three times for 5 min, 100% ethanol twice for 3 min, followed by

royalsocietypublishing.org/journal/rsob Open Biol. 9: 190220

incubation in 95, 80 and 70% ethanol for 1 min each. After haematoxylin and eosin staining, the slides were embedded in TissuePrep-2 Embedding Media (Fisher Scientific), and the images were taken with an AxioObserver Z1 microscope (Zeiss, Germany).

## 2.3. Immunohistochemistry

Staged embryos were fixed in 4% paraformaldehyde overnight at 4°C, cryoprotected with 30% sucrose in phosphate-buffered saline (PBS) and embedded in Optimal Cutting Temperature tissue freezing medium (Triangle Biomedical Sciences, Durham, USA) for cryosectioning. 10 μm (immunohistochemistry) and 16 μm (*in situ* hybridization) transverse sections were collected. For immunohistochemistry, antigen retrieval was performed by boiling the slides in sodium citrate buffer (10 mM sodium citrate and 0.05% Tween 20, pH 6.0) for 20 min in a vegetable steamer. Furthermore, the sections were washed in PBS and incubated for 30 min with an Image iT™ FX signal enhancer (Molecular Probes, USA). Antigen was first recognized by primary antibodies during overnight incubation at 4°C and then visualized using Alexa Fluor 488- or Alexa Fluor 568-conjugated secondary antibodies (Molecular Probes, USA). Cell nuclei were counterstained with DAPI (1 : 1000) for 10 min (Sigma, USA). Slides were washed with PBS and mounted with Vectashield (Vector, USA). Images were taken with a Zeiss Axio Observer fluorescent microscope (Zeiss, Germany). The primary antibodies used: p27$^{Kip1}$ (Santa Cruz Biotechnology, sc-528, 1 : 100), p57$^{Kip2}$ (Abcam, ab75974, 1 : 100), H3K27me3 (Millipore, 07-449, 1 : 100) and SC-35 nuclear speckles (Abcam, ab11826, 1 : 200).

## 2.4. Transcriptome profiling using RNA-Seq

Mouse lenses from E14.5 *Gata3$^{f/f}$*, *Pax6Cre+* (Gata3 null) and *Gata3$^{f/f}$* (control) were dissected out in ice-cold PBS under the microscope. Control and Gata3 null lenses were pulled into three biological replicates; 10 lenses (five embryos) were used for each biological replicate per each genotype. The tissue RNA was isolated using the RNeasy Micro Kit (Qiagen). The lenses were homogenized with an electric pellet pestle motor in 500 μl of Qiazol. Total RNA was purified using micro columns, treated with RNase-free DNase set (Qiagen) and eluted in 14 μl of RNase-free water. The quantity and quality of RNA were analysed using the Qubit fluorometer (Thermo Fisher Scientific) and the Bioanalyzer (Agilent Genomics), respectively. The preparation of strand-specific library was performed using the NEBNext Ultra DNA Library Prep Kit (NEB), and sequencing was run on an Illumina HiSeq 4000 (Novogene, USA).

## 2.5. RNA-Seq data analysis

RNA-Seq reads were aligned to the mouse genome (mm10) using TopHat (v. 2.0.13) [49]. The number of RNA-seq fragments mapped to each gene in the Refseq gene annotation (downloaded from the UCSC genome browser in March 2017) was then counted using HTseq (v. 0.6.1) [50]. Cuffdiff (v. 2.2.1) [51] in the cufflinks package was also used to generate the normalized gene-expression level, fragments per kilobase of transcript per million mapped reads (FPKMs). The RNA-Seq data have been deposited in the Gene Expression Omnibus

(GEO) under accession number GSE131291. In total, 13 024 genes with mean FPKM >1 in either group were detected. Seventy-two differentially expressed genes were determined using the DESeq2 [52] with $p_{adj} < 0.05$ as cutoff. For additional comparisons, RNA-Seq data of the N-Myc [53] and Prox1 [32] lens mutants were compared with Gata3 data to identify commonly dysregulated genes. The Venn diagram and histogram were drawn using the R packages 'Vennerable' and 'upset'. The enriched motifs in the promoters of differentially expressed genes were identified using HOMER (v. 4.7) [54]. The resultant GATA motif predicted was then scanned against promoters of all differentially expressed genes to identify the genes with the GATA motif. The functional analysis of differentially expressed genes was conducted with GSEA (v. 4.0.0) [55].

## 2.6. Quantitative RT-PCR validation

RNA from independent pools of control and mutant lenses was extracted using the RNeasy Micro Kit with on-column DNase digestion (Qiagen). About 1 μg of RNA was reverse transcribed into cDNA using the SuperScript VILO cDNA Synthesis Kit (Thermo Fisher Scientific, USA). The cDNA was diluted 15-fold and qPCR was performed using the StepOne Real-Time PCR System (Thermo Fisher Scientific, USA) and the Power SYBR Green Master mix (Thermo Fisher, USA) in 96-well plates. The relative gene expression was normalized using B2M, HMBS and SHDA reference genes. Student's *t*-test was performed to evaluate differential gene expression between control and Gata3 null lenses. Primers for qPCR are listed in electronic supplementary material, table S1.

## 2.7. Quantitative chromatin immunoprecipitation

Formaldehyde cross-linked chromatin was obtained from a pool of 400 mouse newborn lenses (CD1 mouse; Charles River Laboratories, Cambridge, MA, USA). The sheared chromatin (average size 600 bp of DNA) was generated by sonication [56]. Aliquots of chromatin representing 40 lenses were incubated with 5 μg of anti-Gata3 (Abcam, ab32858), anti-c-jun (Abcam, ab31419), anti-Prox1 (Abcam, ab32858) or anti-Smad1/5/8 (St. Cruz Biotechnology, sc-6331) antibodies bound to 20 μl of protein G-coated magnetic beads (Invitrogen) as we described earlier [36,57]. The immunoprecipitates were washed three times and resuspended in a buffer containing 10 mM Tris–HCl, pH 8.0, 100 mM NaCl, 25 mM EDTA supplemented with 0.1 mg ml$^{-1}$ RNaseA and 0.2 mg ml$^{-1}$ proteinase K. After 2 h incubation at 55°C, the cross-links were reversed by overnight incubation at 65°C. Genomic DNA was eluted into 250 μl of water using the QIAquick Spin Gel Purification kit (Qiagen, Santa Clara, CA, USA). The amounts of each specific DNA fragment in immunoprecipitates were determined by quantitative PCRs using a standard curve generated for each primer set with 0.04, 0.2 and 1% input DNA samples. Using a standard curve, we transformed $C_t$ values into DNA copy numbers. The copy number of a specific DNA fragment in each assay was compared with the copy number of that fragment before immunoprecipitation (input DNA). A control antibody (rabbit normal non-immune IgG from Calbiochem) was included for each set of the qPCR experiments as we described

royalsocietypublishing.org/journal/rsob    Open Biol. 9: 190220

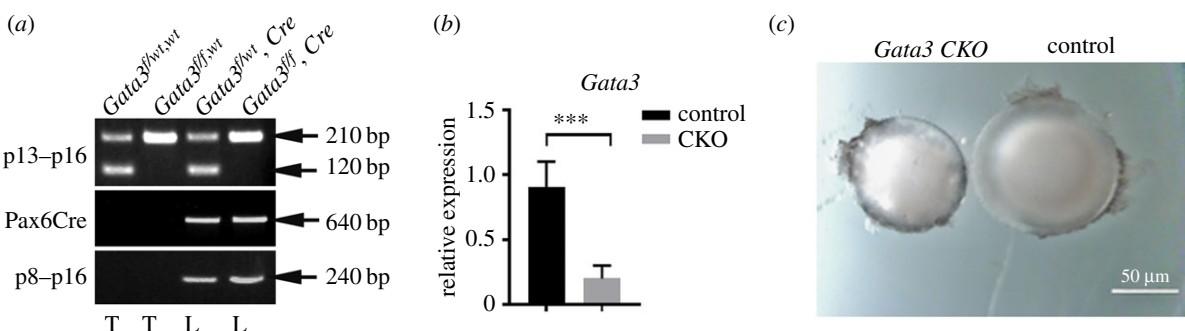

**Figure 1.** Characterization of $Gata3^{L/L}$; *PAX6-Cre* mice. (*a*) Identification of floxed and deleted *Gata3* alleles by PCR after Cre-mediated excision. Samples are from heterozygous floxed (*f/wt*), homozygous floxed (*f/f*), heterozygous floxed (*f,wt*), Pax6-Cre-positive (*f/wt, Cre*) and homozygous deleted Pax6-Cre-positive (*f/f, Cre*) mice (Gata3 CKO). Genomic DNA for genotyping was extracted either from mouse-tail tissue (T) or P0.5 neonatal lens (L). Left margins: primer pairs; right margins: molecular sizes. (*b*) Analysis of Gata3 expression levels in control and *Gata3* null E14,5 lenses by qPCR, $p < 0.01$ (***). (*c*) Comparison of representative lenses from control and *Gata3* mutant (CKO) neonatal mice. Scale bar: 50 μm.

elsewhere [57]. Primers for qPCR are listed in electronic supplementary material, table S2.

# 3. Results

## 3.1. Generation and characterization of lens-specific Gata3 knockout mice

To perform conditional inactivation of *Gata3* in lens cells, we crossed $Gata3^{f/f}$ mice, harbouring loxP sites flanking the *Gata3* exon 4 [46] with Pax6-Cre transgenic mice expressing Cre recombinase in lens and corneal precursor cells of the head embryonic ectoderm as we described elsewhere [47]. Cre recombinase expression in Pax6-Cre mice is controlled by a mouse Pax6 ocular epithelial enhancer along with the P0 Pax6 promoter. PCR analysis of DNA from adult tail and lens tissue, using primers selectively amplifying the unrecombined or the recombined allele, showed that *Gata3* exon 4 was deleted in the lenses of mice expressing Cre (figure 1*a*). qPCR analysis of E14.5 embryonic lens RNA with primers spanning exons 4–5 demonstrated that Gata3 expression was dramatically downregulated (figure 1*b*). Dissection of neonatal mutant eyes revealed reduction in size (figure 1*c*). In addition, opaque spots clustering around the centre of the lens could be readily recognized. Gata3 conditional knockout (CKO) ($Gata3^{f/f}$, Pax6-Cre/+) mice demonstrated eyelid reopening defects compared with control mice: control mice at P15 had open eyelids, while eyelids of *Gata3* CKO mice remained closed (data not shown).

We next performed histological analysis of the Gata3 null lenses at postnatal stages P0 (figure 2*a*) and P15 (figure 2*b*). Mice homozygous for the conditional allele and heterozygous for the Cre transgene were defined as $Gata3^{f/f}$, *Pax6-Cre* (Gata3 CKO thereafter). Heterozygous deletion of *Gata3* ($Gata3^{f/wt}$, *Pax6-Cre*) as well as homo- ($Gata3^{f/f}$, *wt*) and heterozygous ($Gata3^{f/wt}$, *wt*) conditional alleles in the absence of Cre served as experimental controls. The eyes from all control animals showed normal ocular histology (figure 2). Lenticular defects were obvious in the *Gata3* CKO eyes at both P0 and P15 stages (figure 2). Changes within the lens, including fragmentation and liquefaction of lens fibres and swelling of lens epithelial cells (electronic supplementary material, figure S1), are consistent with a cortical cataract. In addition, there was delayed regression of the hyaloid vasculature, which appears

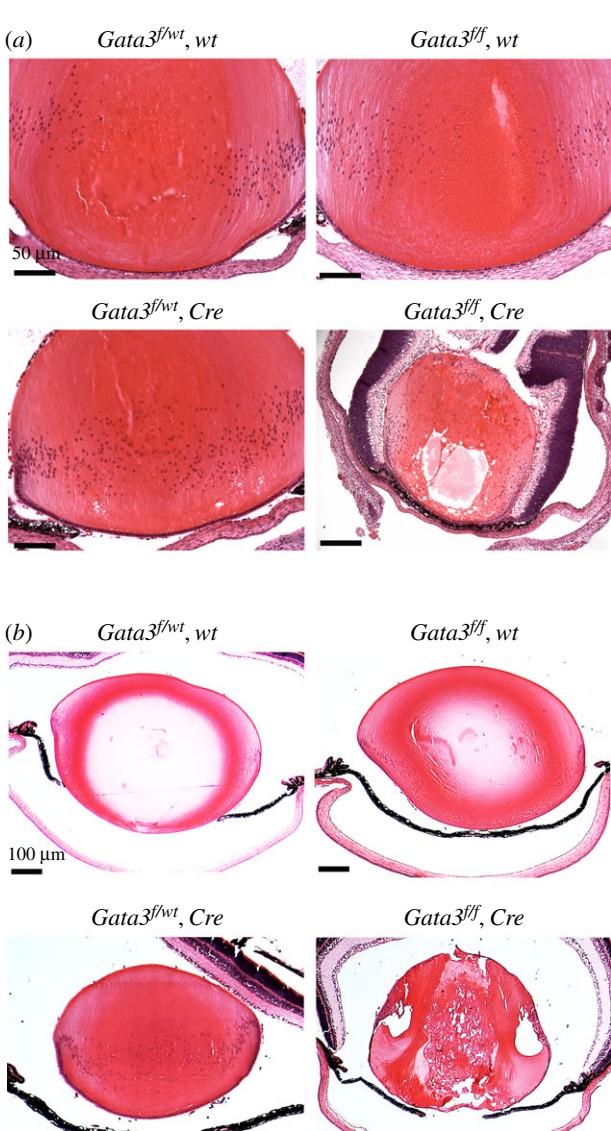

**Figure 2.** Comparative histological analysis of Gata3-depleted lenses with corresponding controls. Haematoxylin and eosin staining of postnatal P0.5 (*a*) and P15 (*b*) mice eyeballs. Note the defects in fibre cell maturation. Scale bars: 200 μm for P0.5 neonatal lens and 400 μm for P15 lens.

to be a secondary mesenchymal developmental defect. The Gata3-depleted lens fibre cells retained their nuclei in the presumptive organelle-free zone (figure 2; electronic

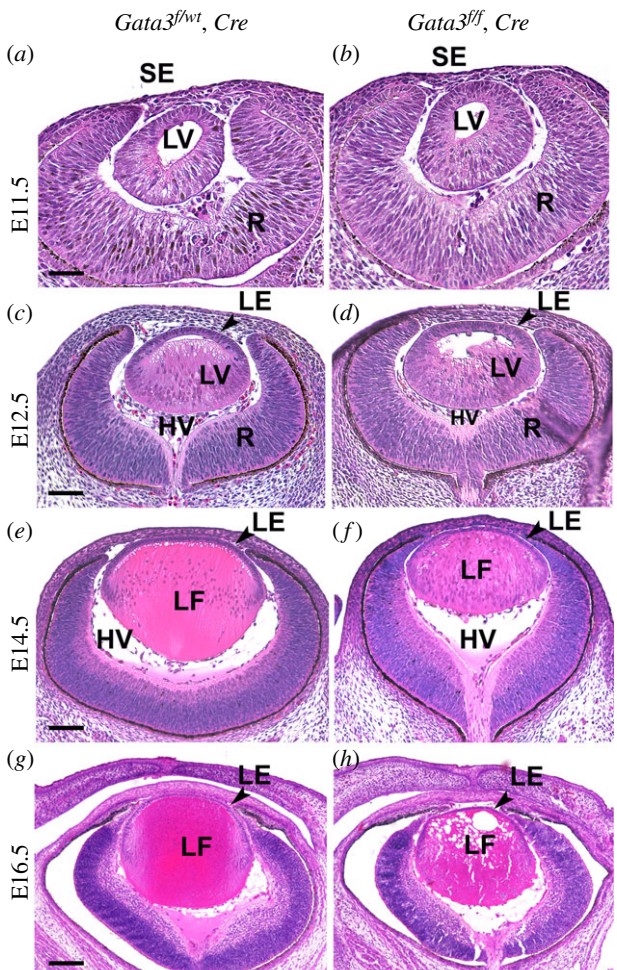

*Gata3^f/wt, Cre*      *Gata3^f/f, Cre*

**Figure 3.** *Gata3* deletion from the early lens compromises primary and secondary fibre cell morphogenesis. (*a*–*h*) Mouse eye sections at E11.5 (*a*,*b*), E12.5 (*c*,*d*), E14.5 (*e*,*f*) and E16.5 (*g*,*h*) stained with haematoxylin and eosin. (*a*,*b*) At E11.5, the lens vesicle has detached from the head surface ectoderm (SE). At this embryonic stage, no apparent differences were detected between Gata3 CKO lens and the corresponding control. (*c*,*d*) Cells at the posterior of the lens vesicle start elongating towards the anterior part of the lens vesicle at E12.5. Primary lens fibre cells in Gata3 null mice exhibit an asymmetric elongation profile compared with the corresponding control. (*e*,*f*) At E14.5, the lumen of the lens vesicle is filled up with elongated lens fibre cells in both Gata3 null and control lenses. (*g*,*h*) At E16.5, Gata3 mutant lenses display defective orientation and elongation of the secondary lens fibre cells as well as denucleation defects. LE, lens epithelium; LF, lens fibres; LV, lens vesicle; HV, hyaloid vasculature; SE, surface ectoderm, R, retina. Scale bars: (*a*–*d*): 100 μm; (*e*–*h*): 200 μm.

supplementary material, figure S1). Altogether, these data demonstrate that conditional inactivation of Gata3 in lens and eyelid precursor cells leads to cortical cataract formation and eyelid opening defects.

## 3.2. Gata3 is required for normal lens development

Previous studies detected Gata3 expression at E10.5 in the lens vesicle [29,58], and our recent findings observed Gata3 expression within the lens and otic pre-placodal regions at E9.5 [37]. The Pax6-Cre line expresses Cre recombinase from E9.5 of lens development [47]. To understand the onset of lens developmental defects, we collected and analysed mouse lenses at different stages of early embryonic development (figure 3). At E11.5, histological examination did not

reveal notable morphological differences between Gata3 CKO and control lenses (figure 3*a*,*b*). Subtle changes in the lenses of the Gata3-deficient mice were first detected at E12.5, as lens fibres displayed slightly asymmetrical elongation patterns (figure 3*c*,*d*). At E14.5, in the control lens, the lens fibre cell nuclei show the regular 'bow-like' pattern; however, the nuclei in mutated lens fibres were positioned posteriorly for the most part (figure 3*e*,*f*). In addition, the fibres themselves were enlarged with finely vacuolated to granular eosinophilic cytoplasm (figure 3*e*,*f*). At E16.5, lens fibre cells were fragmented and replaced by variably sized, irregular, coalescing vacuoles (figure 3*g*,*h*). Taken together, our current study and the earlier studies [29] demonstrate that Gata3 plays an essential role in lens fibre cell differentiation.

## 3.3. Transcriptome analysis of Gata3 mutant and control lenses

To gain additional molecular insights into the role of Gata3 in embryonic lens, we performed mRNA sequencing (RNA-Seq). Since phenotype changes were more prominent at E14.5 (figure 3*f*) compared with E12.5 (figure 3*d*), we collected homozygous mutant (*Gata3^f/f, Pax6-Cre*) and control (*Gata3^f/wt, wt*) lenses at this developmental stage. We pooled 10 lenses from five embryos and used three distinct pools as biological replicates for mutant (CKO) and control (WT) samples. The mRNA libraries were paired-end sequenced on an Illumina HiSeq 4000 and subsequently analysed as described in Material and Methods. The libraries generated a total of 144 million paired-end reads (electronic supplementary material, figure S2a). Excision of Gata3 exon 4 in mutant lenses was confirmed by RNA-Seq (electronic supplementary material, figure S2b). In total, we identified 13 024 genes expressed in both samples. Importantly, 72 significantly differentially expressed genes at adjusted $p$-values ($p_{adj}$) $< 0.05$ were detected in *Gata3* null lenses (figure 4*a*; electronic supplementary material, figure S3a). Within this group, 43 and 29 genes were downregulated and upregulated, respectively (electronic supplementary material, table S3). RNA-Seq results were subsequently confirmed by qPCR using independent pools of mutant and control lenses with merged results from three biological replicates for upregulated and downregulated genes as shown in figure 4*b*,*c*. The list of most downregulated mRNAs includes Pla2g7, Pgam2, Stx11, Myo7b, Trpc6, Dnase2b, Nlrc5, Lenep, Pde2a, Scnn1b, Bfsp2, Optn and Hopx (electronic supplementary material, table S3). *Dnase2b* encodes the acidic DNase IIβ enzyme required for fibre cell denucleation [59]. Downregulation of lens-specific intermediate filament protein CP49/phakinin (Bfsp2) can explain abnormal lens fibre cell elongation and lens morphogenesis [60,61]. Interestingly, *Bfsp2* and *Lenep* (LEP503, [62]) rank within top 30 most highly expressed genes at E14.5 lens fibre cells [63]. In addition, through visualization in the iSyTE2.0 database, all these 13 genes are markedly upregulated in lens between E12.5 and E16.5 [64]. The most upregulated mRNAs in Gata3 CKO lenses are Nptx2, Meg3, Sstr1, Ednrb, Dcdc2a, Eva1a, Ryr3, Eda2r, Rian, Vstm2b, Pcp4l1, Atp10d and Megf10. Within this group, Ednrb and Meg3 show much higher lens expression compared with the remaining 11 genes [64]. Endothelin receptor type B (*Ednrb*) is a G-protein coupled receptor implicated in calcium signalling, and *Meg3* encodes paternally imprinted long noncoding RNA. Noteworthy, the expression of several

royalsocietypublishing.org/journal/rsob   Open Biol. 9: 190220

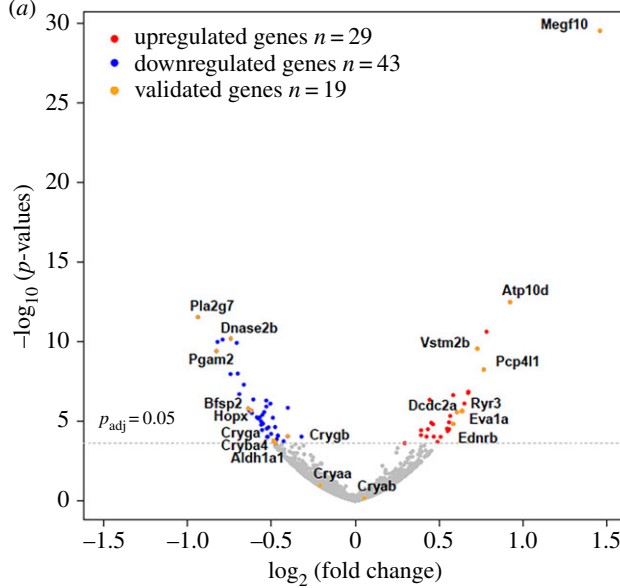

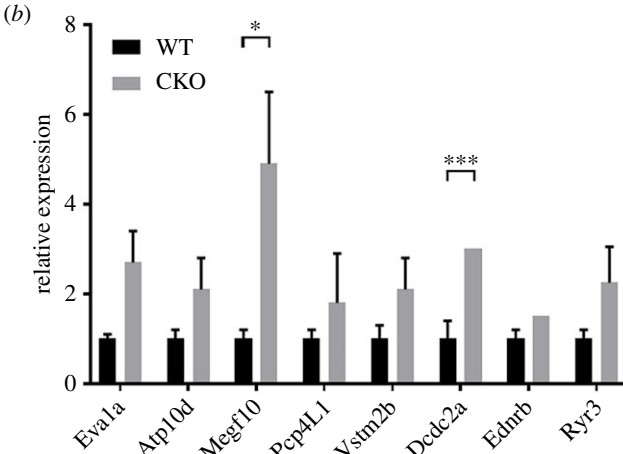

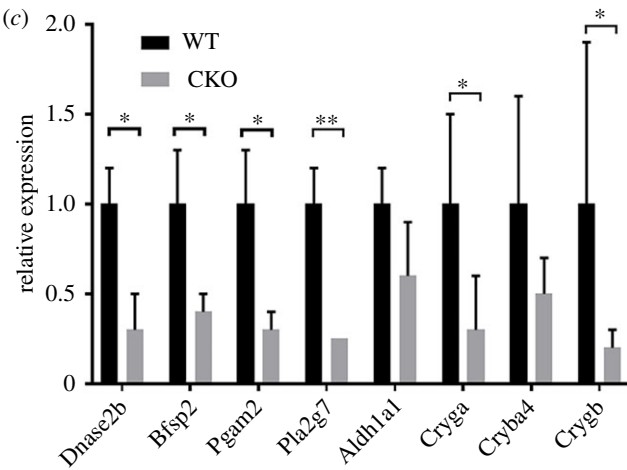

**Figure 4.** Transcriptomic analysis of *Gata3* mutant lenses. (*a*) Volcano plot of differentially expressed genes. Red and blue points mark the genes with increased or decreased expression, respectively, in Gata3 CKO samples compared with *Gata3* controls. Orange points correspond to genes that were further validated by qPCR. The *x*-axis and *y*-axis show $\log_2$ fold changes and *p*-values of a gene being differentially expressed, respectively. (*b,c*) Validation of RNA-Seq data for differential gene expression by qRT-PCR. qRT-PCR analysis was performed on RNAs from *Gata3* control and *Gata3* mutant lenses using the primers listed in electronic supplementary material, table S2 as described in Material and Methods. The data are representative of three independent experiments performed on three biological replicates; mean ± s.e.m.

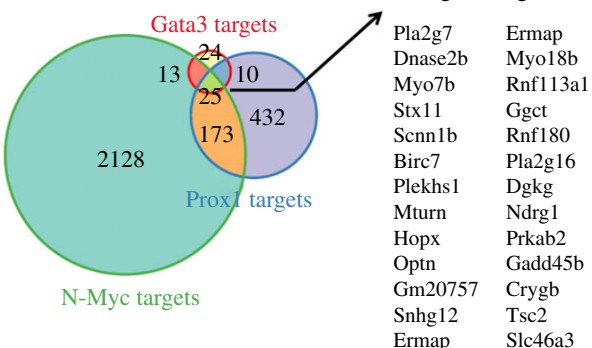

**25 common downregulated genes:**

| | |
|---|---|
| Pla2g7 | Ermap |
| Dnase2b | Myo18b |
| Myo7b | Rnf113a1 |
| Stx11 | Ggct |
| Scnn1b | Rnf180 |
| Birc7 | Pla2g16 |
| Plekhs1 | Dgkg |
| Mturn | Ndrg1 |
| Hopx | Prkab2 |
| Optn | Gadd45b |
| Gm20757 | Crygb |
| Snhg12 | Tsc2 |
| Ermap | Slc46a3 |

**Figure 5.** Comparative analysis of genes with altered expression in Gata3, N-Myc and Prox1 lens loss-of-function models. The differentially expressed genes of Prox1 and N-Myc knockout were obtained from the previously published papers [32,53] and compared with the targets of Gata3 knockout.

crystallin genes (*Crygb*, *Cryga* and *Cryba4*) was downregulated in *Gata3* mutant lenses (figure 4*c*), consistent with abnormal differentiation. Most of the differentially expressed genes exhibited changes of less than twofold, but there is evidence that modest changes in RNA levels can be biologically relevant [65]. A number of the differentially expressed genes contain Gata3-binding sites in their putative regulatory regions (electronic supplementary material, figure S4).

To gain additional insights into these processes, we compared Gata3-regulated genes with altered expression genes from similar loss-of-function studies of transcription factors N-Myc [53] and Prox1 [32] analysed at comparable developmental stages by RNA-Seq (figure 5). The comparison revealed 25 commonly downregulated genes, including *Dnase2b* and *Crygb*, discussed above. By contrast, no common upregulated gene was identified. The individual genes found are summarized in table 1 and include only a few examined so far in lens, e.g. *Dnase2b*, *Birc7*, *Hopx* and *Crygb*. Other genes represent solid candidates for follow-up studies of autophagy (*Gadd45b*), cell polarity (*Ndrg1*), energy metabolism (*Prkab2* and *Tsc2*), intracellular signalling (*Plekhs1*) and oxidative stress (*Ggct* and *Pla2g7*). Taken together, we conclude that Gata3 functions as both an activator and repressor of transcription in lens cells, similar to what was shown earlier in other tissues [83–87], the resulting lens abnormalities originate from both up- and downregulated genes, and a small number commonly dysregulated genes in *Gata3*, *N-Myc* and *Prox1* lens genetic models exist indicating both common and unique pathways under their genetic control.

## 3.4. Downregulation of Cdk inhibitors p27^Kip1 and p57^Kip2

Maeda *et al*. [29] previously demonstrated downregulation of Cdkn1b/p27 and Cdkn1c/p57 expression in TghDBH-G3-rescued Gata3 null mutant lens. Here, we found that both Cdkn1b/p27 and Cdkn1c/p57 proteins were downregulated in the transition zone in Gata3-depleted lenses (figure 6). To understand how Gata3, Cdkn1b/p27 and Cdkn1c/p57 are regulated by DNA-binding transcription factors, we used quantitative chromatin immunoprecipitation (qChIP) assays. Previously, we identified a bipartite 5′-located 1AB enhancer of the Gata3 locus [36]. The current data show that c-Jun and

royalsocietypublishing.org/journal/rsob    Open Biol. 9: 190220

**Table 1.** Twenty-five genes commonly downregulated in Gata3, N-Myc and Prox1 lens loss-of-function studies (analysed by RNA-Seq[a]).

| gene symbol | protein name | structure and function |
| --- | --- | --- |
| Pla2g7 | Phospholipase A2, group VII | cytoplasmic enzyme is implicated in chronic inflammation and oxidative stress |
| Dnase2b | Acidic DNase IIβ | key enzyme to degrade DNA during lens fibre cell denucleation [15,59] |
| Myo7b | Myosin VIIB | unconventional myosin with tissue-restricted expression |
| Stx11 | Syntaxin 11 | syntaxins are a family of nervous system receptors implicated in the docking of synaptic vesicles with the presynaptic plasma membrane |
| Scnn1b | Sodium channel, nonvoltage-gated 1 beta | multipass membrane protein and cytoplasmic vesicle membrane protein |
| Birc7 | Baculoviral IAP repeat-containing 7 (livin) | a member of the inhibitor of apoptosis protein (IAP) family is strongly upregulated in deep cortical fibre cells; however, loss of function of Birc7 produces normal lenses [66]; it contains RING domain protein with putative E3 ubiquitin ligase activity |
| Plekhs1 | Pleckstrin homology domain containing, family S member 1 | Pleckstrin homology domain proteins are involved in intracellular signalling or within cytoskeleton; tissue-restricted expression or unknown function |
| Mturn | Maturin, neural progenitor differentiation regulator homologue | novel regulator of early neurogenesis including retinogenesis [67] |
| Hopx | HOP homeobox | the smallest known member of the homeodomain-containing protein family, though unable to bind DNA, regulates neocortex [68] and highly expressed in maturing lens fibres [69] |
| Optn | Optineurin | an ubiquitin-binding scaffold protein; multifunctional protein that functions as an important autophagy and the mitophagy receptor in selective autophagy processes and as a cell cycle regulator [70] |
| Gm20757 | Predicted gene, 20757 | ncRNA on mouse chromosome 10, function: unknown |
| Snhg12 | Small nucleolar RNA host gene 12 | ncRNA on mouse chromosome 4, function: unknown |
| Ermap | Erythroblast membrane-associated protein | composed from a transmembrane segment and one extracellular immunoglobulin fold (IgV); its cytoplasmic domain contains a highly conserved B30.2 motif and post-Golgi sorting signals; function: unknown |
| Myo18b | Myosin XVIIIb | unconventional myosin with tissue-restricted expression in the heart [71] |
| Rnf113a1 | Ring finger protein 113A1 | potential role in splicing |
| Ggct | γ-Glutamyl cyclotransferase | implicated in glutathione homeostasis and modulates a bipolar cell number in retina [72] |
| Rnf180 | Ring finger protein 180 (rines) | Rines is a membrane-bound E3 ubiquitin ligase [73] |
| Pla2g16 (Plaat) | Phospholipase A and acyltransferase 3 | a phospholipase catalyses phosphatidic acid into lysophosphatidic acid and free fatty acid implicated in cancer metastasis [74] |
| Dgkg | Diacylglycerol kinase, gamma | this tissue-restricted kinase (mostly the CNS) is an upstream suppressor of Rac1 and, consequently, lamellipodium/ruffle formation [75] |
| Ndrg1 | N-myc downstream regulated gene 1 | cytoplasmic protein required for neuronal polarity [76] |
| Prkab2 | Protein kinase, AMP-activated, beta 2 non-catalytic subunit | non-catalytic subunit of AMP-activated protein kinase (AMPK); AMPK is an energy sensor protein kinase that plays a key role in regulating cellular energy metabolism [77] |
| Gadd45b | Growth arrest and DNA damage-inducible 45 beta | a 160 aa ubiquitous nuclear fasting and stress-induced protein involved in the regulation of growth and apoptosis [78] and the negative regulator of autophagy [79] |
| Crygb | γB-Crystallin | a major lens structural protein [80,81] |
| Tsc2 | TSC complex subunit 2 (tuberin) | Tuberin (a large 1814 aa phosphoprotein), in complex with TSC1, inhibits the nutrient-mediated or growth factor-stimulated phosphorylation of S6K1 and EIF4EBP1 by negatively regulating mTORC1 signalling [82] |
| Slc46a3 | Solute carrier family 46, member 3 | multi-pass membrane protein (460 aa) |

[a]References only included if there is any possible connection to known lens biology.

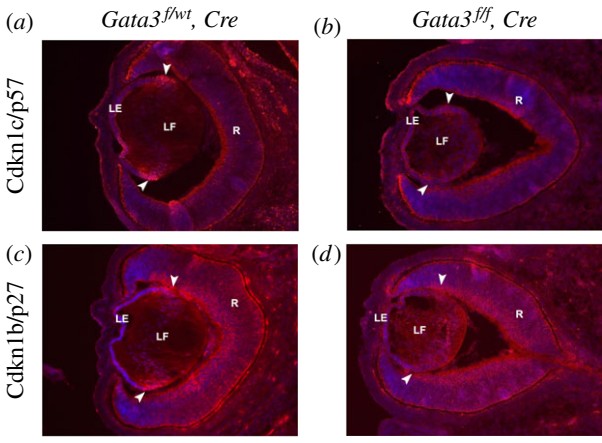

*(a)* Gata3*^{f/wt}*, Cre    *(b)* Gata3*^{f/f}*, Cre

Cdkn1c/p57

Cdkn1b/p27

*(c)*    *(d)*

**Figure 6.** Dysregulation of Cdkn1c/p57 Cdkn1b/p27 expression in *Gata3^{−/−}* lenses. Immunohistochemical analysis shows elevated Cdkn1b/p27 and Cdkn1c/p57 protein levels (arrowheads) at the equatorial region of the E14.5 WT lens (*a,c*), whereas reduced expression is observed in mutant littermates (*b,d*). The nuclei were counterstained with DAPI (blue). LE, lens epithelium; R, retina; LF, lens fibres.

Prox1 bind *in vivo* to DNA overlapping the enhancer 1AB in lens chromatin prepared from newborn lenses while c-Jun, Prox1 and Smad1/5/8 recognize another 3'-located evolutionarily conserved enhancer (figure 7*a*). Smad-binding site was previously described at −4.3 kb region of the Gata3 promoter using chromatin from bone cells [88]. Importantly, qChIP analyses of 5'-flanking regions of *Cdkn1b/p27* and *Cdkn1c/p57* loci in lens chromatin prepared from newborn lenses revealed the presence of Gata3 as well as other DNA-binding transcription factors, including Prox1, c-Jun and Smad1/5/8 (figure 7).

## 3.5. Disrupted subnuclear organization in Gata3 CKO lens fibre cells

The molecular mechanisms underlying lens fibre cell denucleation remain poorly understood [15,16,89]. Retention of nuclei in *Gata3*-mutated lens fibres could be accompanied by additional defects such as abnormalities in modified core histone subunits implicated by our recent studies analysing chromatin defects in the stalled lens fibre cell denucleation process [17]. To examine this possibility, we used antibodies to visualize H3K27me3 in the lens fibre cell nuclei of E14.5 embryos. H3K27me3 chromatin domains are directly associated with transcriptional repression [90]. We found that Gata3-depleted lenses display increased signals of this modified histone compared with the control nuclei (figure 8). To extend these studies, we tested antibodies that recognize RNA speckles, also called interchromatin granule clusters. These structures represent nuclear domains containing pre-mRNA splicing factors and reflect functional organization of the nucleus [91]. We found marked reduction of nuclear speckles in the Gata3 CKO lenses (figure 8). Taken together, these data show perturbed subnuclear organization of Gata3-depleted lens fibre cells even prior full execution of the denucleation process.

## 4. Discussion

One goal of the present study was to determine whether lens-specific inactivation of Gata3 was sufficient to generate defects in lens fibre cell differentiation. A second goal was to further define the molecular functions of Gata3 in the embryonic lens. Using unbiased transcriptomic analysis, we show that Gata3 regulates a relatively small fraction of the lens transcriptome compared with other lens transcription factors, including N-Myc [53], Prox1 [32] and Zeb2 [92] (electronic supplementary material, figure S5). We confirm the reduction of Cdkn1b/p27 and Cdkn1c/p57 expression in Gata3-mutated lenses and demonstrate *in vivo* binding of Gata3 to putative upstream regulatory sequences for both genes. We also show that other proteins implicated in cell cycle exit in lens cells, including c-Jun, Prox1 and Smad1/5/8, bind to these loci. Our data also predict that Prox1 regulates directly Gata3 expression. A summary model of these interactions is shown in figure 9. Depletion of Gata3 causes abnormal fibre cell maturation and retention of the nuclei in the prospective organelle-free zone of maturing fibre cells, and these retained nuclei show abnormal morphology and internal organization.

Earlier functional studies of Gata3 in lens employed a somatic null mutant rescued by transgenic expression of Gata3 and established for the first time expression of Gata3 proteins in the posterior part of the WT lens vesicle [29]. While the somatic loss/rescue strategy inactivates Gata3 earlier and more broadly compared with Pax6-Cre, both studies identified lens developmental abnormalities that originated at approximately E12.5. It is relevant to note that Cre recombinase in the Pax6-Cre line is expressed in scattered cells within the retinal neuroectoderm [47], but previous as well as our recent studies [37] did not detect any significant Gata3 expression in retina [58,93,94]. In concordance with this, the retina of the conditional knockout is within normal limits (figures 2 and 3). Using RNA-Seq, the present study identified downregulated and upregulated genes in the Gata3 CKO lens. In the group of 72 dysregulated genes (electronic supplementary material, table S3), upregulation of Eda2r mRNAs was found here as well as in the analysis of differentially expressed genes in le-cre hemizygous lenses [48]. Further studies are needed to test if Pax6-cre transgene may elicit subtle gene expression changes as a result of transgene integration to be included as additional controls for RNA-Seq studies. Notably, downregulated genes encode three crystallin proteins (γA-, γB and βA4-crystallins), intermediate filament protein Bfsp2 and acidic DNase IIβ. The other downregulated Pla2g7, Pgam2, Stx11, Myo7b, Trpc6, Nlrc5, Lenep, Pde2a and Scnn1b mRNAs encode proteins with unknown roles in the lens. The upregulated Nptx2, Meg3, Sstr1, Ednrb, Dcdc2a, Eva1a, Ryr3, Eda2r, Rian, Vstm2b, Pcp4l1, Atp10d and Megf10 genes, currently unknown in lens biology, can also contribute to the lens abnormalities found here. These genes may prove useful for comparative studies involving loss-of-function studies of other regulatory genes in the lens. Regarding the commonly downregulated genes in Gata3, N-Myc and Prox1 lens-specific loss-of-function studies, the most parsimonious interpretation is that they represent a small group of co-regulated genes under either joint direct or indirect controls. Alternatively, it is possible that their dysregulation reflects a signature of immature lens fibre cells.

Our studies confirm downregulation of Cdkn1b/p27 and Cdkn1c/p57 in lens at the protein level as shown earlier [29] and support their direct regulation by both Gata3, perhaps in conjunction with Prox1. Nevertheless, our bulk RNA-Seq

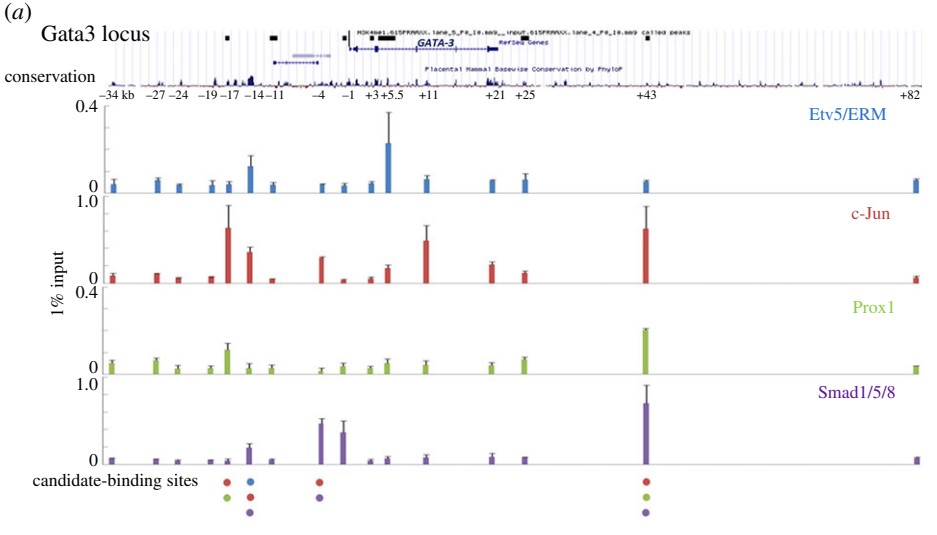

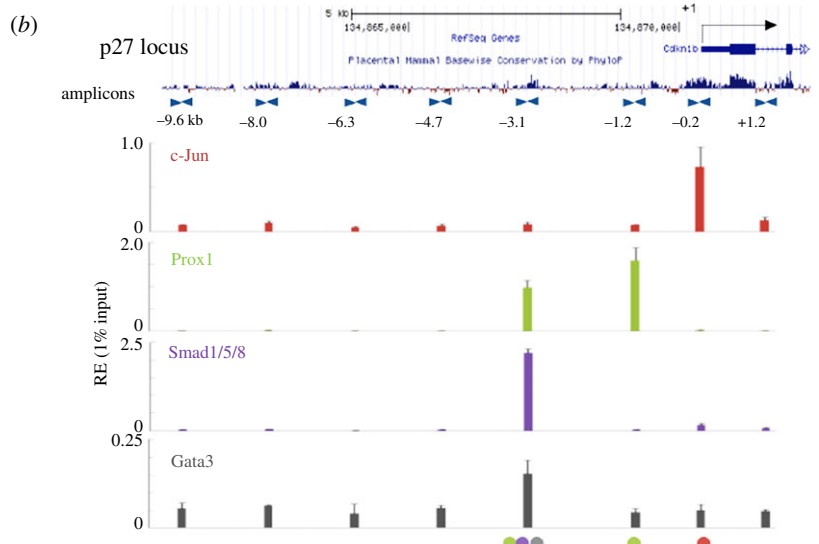

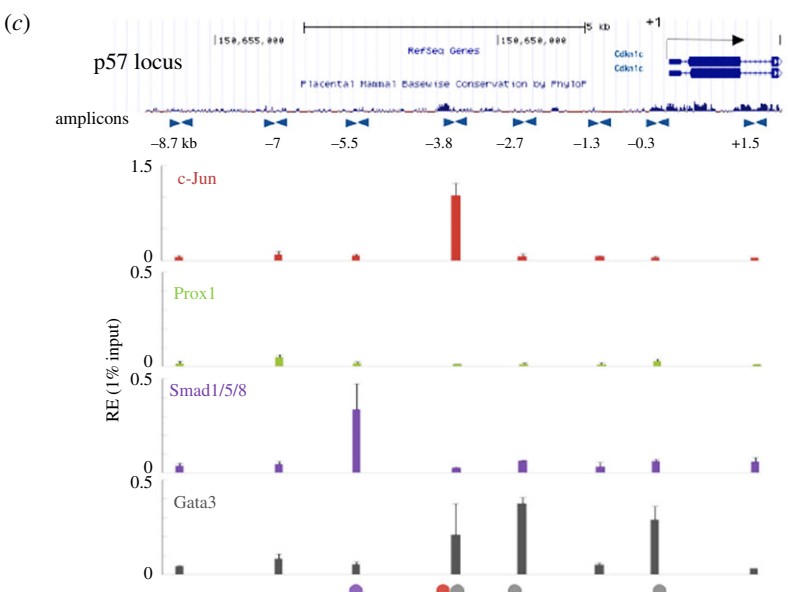

**Figure 7.** Expression of Cdkn1b/p27 and Cdkn1c/p57 in lens is regulated by Gata3 at the level of transcription. (*a*) Localization of c-Jun, Prox1, Smad1/5/8 and Etv5/ERM determined by qChIP analysis over the 120 kb Gata3 locus in lens chromatin. The relative enrichments are shown as 1% of the input. (*b*) Analysis of the mouse p27 locus. Binding of Prox1, Smad1/5/8 and Gata3 was observed at the evolutionary conserved −3.1 kb 5′-promoter region. (*c*) Analysis of the mouse p57 locus. Binding of Gata3 and c-Jun was detected at the evolutionary conserved −3.8 kb 5′-promoter region. No Prox1 binding was found within the 12 kb region analysed. Localization (red-, orange-, purple- and green-coloured filled circles) of these DNA-binding factors was mostly detected in evolutionary conserved non-coding regions p27: −3.1 kb, −1.2 kb and promoter and p57: −5.5 kb, −3.8 kb and promoter.

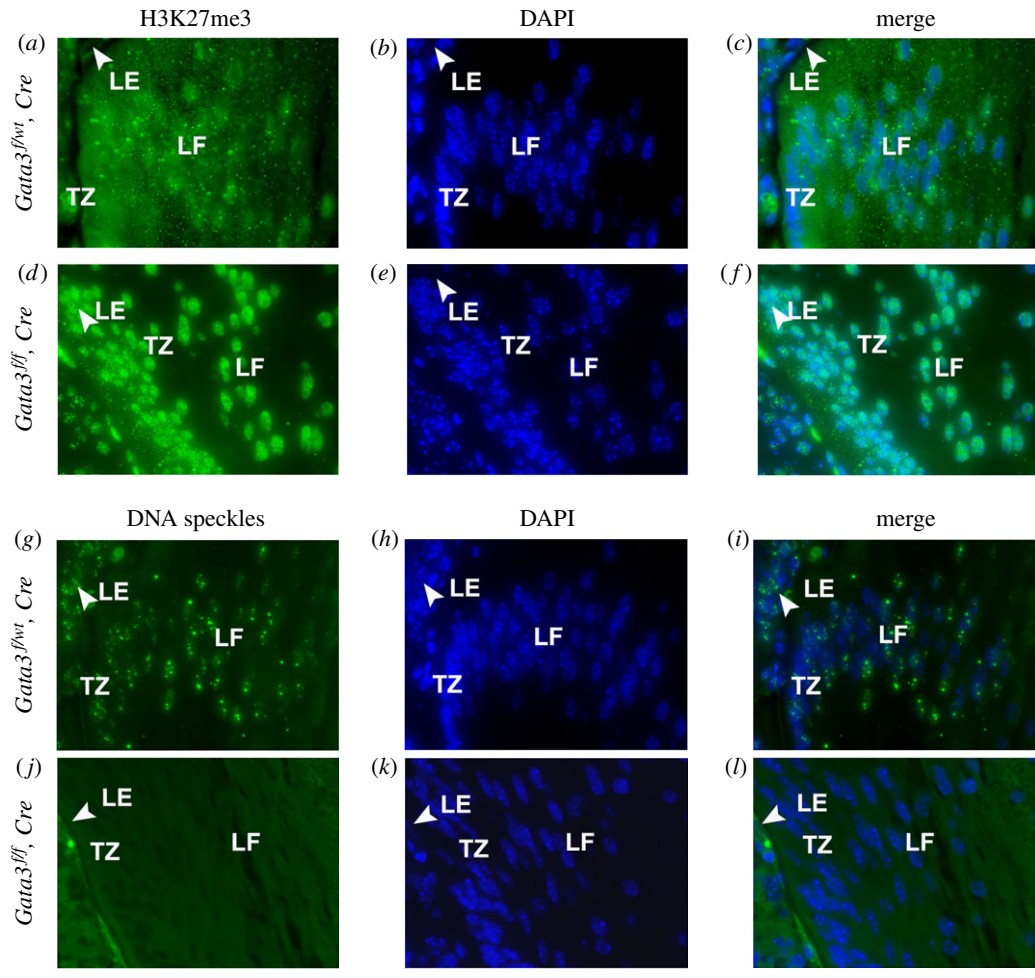

**Figure 8.** Inactivation of Gata3 leads to increased H3K27me3 and elimination of the nuclear speckles. The level and distribution of H3K27me3 (*a–f*) and SC35 (*g–l*) in the lens fibre cells at the equatorial region of E14.5 wild-type and *Gata3*$^{-/-}$ mouse lens was examined by indirect immunofluorescence microscopy. Loss of *Gata3* resulted in an increase in the levels of H3K27me3 (*a–c* versus *d–f*). Nuclear speckle domains in the cells at the equatorial region of the mouse lens are SC35-positive (*g–i*). Gata3 knockout resulted in the elimination of the SC35-positive speckle domains (*j–l*). Nuclei were visualized using DAPI. Magnification, ×40.

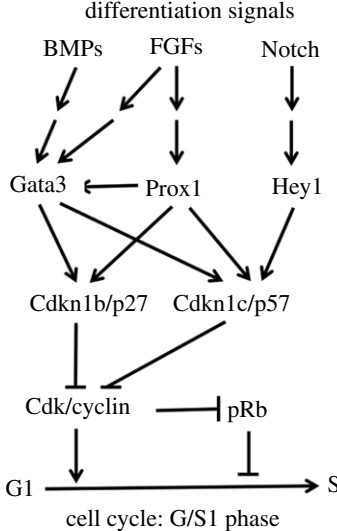

**Figure 9.** A summary diagram of FGF, BMP and Notch signalling-regulated transcriptional mechanisms that govern the cell cycle exit pathways of lens cells, mediated by p27/p57 proteins, and required for lens fibre differentiation.

studies at E14.5 did not identified a change in p27 and p57 mRNAs. Additional studies are needed to examine expression of these mRNAs using high-resolution RNA *in situ* hybridizations at multiple stages between E12 and E16.5 developmental stages, and consider posttranscriptional regulatory mechanisms that regulate the stability of p27 and p57 mRNAs [95,96]. In fact, a recent study has shown that the RNA-binding protein Celf1 binds to GU-rich sequences in the 5′-UTR of p27 mRNA to regulate its stability in lens [97]. Earlier studies of Prox1 in lens also demonstrated that Prox1 was genetically upstream of both Cdkn1b/p27 and Cdkn1c/p57 [31,32]. Hey1/Herp2, a nuclear target of Notch signalling, was also found to regulate directly the Cdkn1c/p57 promoter in lens cells [98]. Earlier studies have shown that Med1/PBP, a peroxisome proliferator-activated receptor (PPAR)-binding protein and co-activator of Gata3, is essential for lens formation [99]. However, it remains to be established whether Med1 and Gata3 work together in the lens as significant lens abnormalities were found in Med1 null lenses already at E11.5 [99].

Lens fibre cell denucleation is a complex process that culminates in generation of the organelle-free zone around E18.5 of mouse embryonic development [15,16,89]. Two key molecular mechanisms involve lamin B phosphorylation by Cdk1 to disrupt the nuclear lamina [100] and lens-specific upregulation of Dnase2b and concentration of the DNase IIβ enzymes in vesicles that attach outside of the nuclei [59]. Cell culture experiments identified two DNA-binding transcription factors, Hsf4 and Pax6, as direct regulators of Dnase2b promoter [101,102]. Our data also predict Gata3 candidate-binding sites in the proximal DNase IIβ promoter (electronic supplementary material, figure S4).

royalsocietypublishing.org/journal/rsob Open Biol. 9: 190220

The findings of increased signals H3K27me3 and decreased nuclear speckles in lens fibre cell nuclei of Gata3-depleted lenses were quite unexpected. Previous studies have shown that H3K27me3 lens nuclear signals are abundant in ovoid shape nuclei; however, during their condensation in maturing lens fibres, reduction of staining was detected [17]. This reduction is caused by transfer of these histones into the cytoplasm, a process originally discovered in maturing mouse erythrocytes prior their enucleation [103]. By contrast, the present studies show increased H3K27me3 stainings in the Gata3 null lens fibre cell nuclei consistent with the arrested denucleation processes. These patterns cannot be readily explained by transcriptomic data and will require follow-up studies, particularly during lens fibre cell denucleation. Likewise, marked reduction of DNA speckles in mutated lens fibre cell nuclei indicates disrupted subnuclear organization implicating Gata3 proteins as important regulators of general nuclear architecture. Studies of splicing in mammalian lenses remain in their infancy [104], and the current datasets provide opportunities for future comparative studies. Future experiments are needed to examine mutated lens fibre cell nuclei using additional modified histones, components of mRNA splicing and through analysis of the nucleolar structure and function.

## 5. Conclusion

Taken together, the present studies provide additional new insights into the molecular mechanisms of lens development regulated by transcription factor Gata3 and identify Gata3 as a potential direct regulator of Cdkn1b/p27 and Cdkn1c/p57

expression during primary lens fibre cell differentiation. In addition, Bfsp2 and DNase IIβ are essential proteins for lens fibre cell differentiation downstream of Gata3. Overall, lens abnormalities found in the Gata3 model are less severe compared with the loss-of-function studies of other transcription factors, including N-Myc and Prox1, consistent with the different numbers of differentially expressed genes in each of these three models. Finally, Gata3-depleted nuclei exhibit major morphological changes that pave the road for follow-up experiments related to the degradation of nuclei in maturing lens fibre cells.

Ethics. Animal husbandry and experiments were conducted in accordance with the approved protocol of the Albert Einstein College of Medicine Animal Institute Committee and the ARVO Statement for the Use of Animals in Ophthalmic and Vision Research.

Data accessibility. The RNA-Seq data are available from the GEO under accession number GSE131291.

Authors' contributions. E.M. conducted the majority of experiments and their analyses and drafted the manuscript. Y.Z. and D.Z. performed all the bioinformatics analyses. Q.X. generated the qChIP data. A.C. conceived the study and finalized the manuscript.

Competing interests. The authors declare that they do not have a conflict of interest.

Funding. The project was funded by NIH/NEI (grant no. R01 EY012200 to A.C.).

Acknowledgements. We thank Dr Amanda P. Beck from the Histotechnology and Comparative Pathology Facility at the Albert Einstein College of Medicine for help with phenotype description. We are grateful to Dr Paul A. Overbeek for the Pax6-cre mice and critical reading of the manuscript. We are also grateful to Dr Jinfang Zhu (NIAID/NIH) for the Gata3$^{Flox}$ mouse model. We thank Ms Jie Zhao and Dr Rebecca McGreal-Estrada for help with the mice colonies.

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
