## [Reviewer comments · Open Biology]

Review History

RSOB-19-0220.R0 (Original submission)

Review form: Reviewer 1

Recommendation

Major revision is needed (please make suggestions in comments)

Do you have any ethical concerns with this paper?

No

Comments to the Author

Martynova et al. explore the role of Gata3 in lens development using a tissue-specific knockout approach. They confirm published results that Gata3 is required for proper lens differentiation and that loss of Gata3 results in congenital cataract. They perform RNAseq and show that some putative Gata3 target genes are also altered in other models of congenital cataract mutants. They also confirm the published reductions in p27/cdkn1b and p57/cdkn1c in Gata3 knockout lenses, arguing that the effect may be direct. Finally they report changes in the chromatin organization of Gata3 knockout lenses.

The work is mostly well-done and an important confirmatory finding about Gata3 in the lens.

The RNAseq data may add important mechanistic information about what functions of Gata3 are essential, although this is outside the body of this work. There are several major and minor concerns for this publication that should be addressed prior to the publication of the work:

- 1) The authors correctly utilize a number of controls to demonstrate a lens phenotype, including those containing the Cre transgene. Since expression of Cre itself can be toxic to lens cells (see Lam PT, et al. Hum Genomics. 2019 Feb 15;13(1):10), all phenotypic work in the lens should be done with a Cre/+ control, which is shown in Fig. 1 and 2. However other important experiments like RNAseq were performed in Cre- controls, despite the known transcriptional changes associated with Cre-expression in the lens.
- 2) The control lenses in Fig. 1 are too small and low-resolution to properly assess the nuclear structure. This is a concern for the Cre/+; Gata3fl/+ lens, which appear to show nuclear retention, although this may also be due to utilizing a peripheral section. This figure should be enhanced so that nuclear structure can be properly assessed in appropriate central sections from all genotypes.
- 3) The immunofluorescent data for p27/cdkn1b and p57/cdkn1c are suggestive of reductions in protein levels, but the overall evidence presented for these proteins is too low to be conclusive. Several points need to be addressed:
 - i) If the authors wish to make the point that this is potentially transcriptional control, they need to determine why the mRNA levels for these genes are not decreased, at least as far as their RNAseq criteria go.
 - ii) Much of the reported regulation of p27/cdkn1b and p57/cdkn1c occurs post-translationally, which is not discussed in the paper. It would not be surprising if the observed reduction in protein is due to altered protein stability. This needs to be addressed, at minimum utilizing RNA expression versus protein levels. Preferably, experiments should be performed in lens cell culture models using Gata3 knockdown/CRISPR approaches and modulation of protein stability via proteasome inhibition.
 - iii) The overall presented immunofluorescence data is relatively low quality and low resolution. These images need to be enhanced, either using higher magnification, or better microscopy. Furthermore, some of the antibodies appear to have high background or non-specific staining in other tissues.
- 4) The common phenotypic results of Gata3, Prox1, and N-Myc knockout lenses and their partially overlapping transcriptomic changes can be due to functional relatedness, as proposed by the authors and their model in Figure 8, but can also be due to overlapping indirect developmental stunting. In this respect, the transcriptomic data is indicating a signature of an immature or blocked lens fiber cell. This can be partially assessed using a different kind of cataract that is caused by a mutation in a structural protein, like beta crystalline, rather than a transcriptional regulator. At minimum, the authors need to expand their discussion of potentially indirect findings.
- 5) It is unclear why chromatin IP analysis is being performed at P0, while the RNAseq data and immunofluorescence analysis for p27/cdkn1b and p57/cdkn1c was performed at E14.5. In the case of p27/cdkn1b and p57/cdkn1c, the protein staining should be performed at P0 as well as repeated at E14.5. If there are differences, the ChIP may need to be repeated at E14.5
- 6) The authors probe potential binding sites for Gata3 upstream of cdkn1b and cdkn1c. Is it clear that these are the relevant regulatory regions that control expression of these genes in the lens? Furthermore, what is the overall frequency of Gata3 sites within the genome, and are these sites likely to be by chance or because of specific enrichment. Because there are not transcriptional changes associated with these binding sites, the data is less compelling.
- 7) The data evaluating H3K27 tri-methylation and nuclear speckles is highly phenomenological and does not advance any specific hypothesis. Given that these may be markers of a developmentally blocked lens, it would be useful to evaluate in a different congenital cataract model to assess whether this is specific to Gata3 inactivation or not.
- 8) There are minor grammatical errors throughout the manuscript (e.g. page 10, line 184 and 187).

Review form: Reviewer 2

Recommendation

Accept with minor revision (please list in comments)

Do you have any ethical concerns with this paper?

No

Comments to the Author

The manuscript by Martynova et al. focuses on characterization of lens defects on the phenotypic and molecular level in *Gata3* (encoding a transcription factor) lens-specific conditional knockout mice. The motivation of this study was to understand *Gata3* function in the lens. This was based on the expression pattern of *Gata3* in the developing lens as well as a previous study describing a rescue experiment in *Gata3* germline mice that still showed the lack of rescue of lens defects, in turn, suggesting its potential function in this tissue. The manuscript describes that *Gata3* cKO lenses exhibit nuclear degradation defects and identifies 72 differentially expressed RNAs by RNA-seq. Furthermore, by comparative analysis with other RNA-seq data from defective lenses of various TF-deletion mice (e.g. *Prox1*, etc.), they identify 25 commonly regulated candidates. Additionally, by qChIP analyses, the authors uncover p27 and p57 as potential direct targets of *Gata3* in the lens through the identification of *Gata3* occupancy in their potential regulatory regions. Overall, the manuscript is very well written and the data are clearly presented and provide a significant advance in the understanding of the role of *Gata3* in autonomously controlling gene expression regulation in fiber cell differentiation during lens development. Further, these data identify several new targets of *Gata3*, with many of these targets representing new potential candidates that are excellent for further investigation in the lens. Because *Gata3* is an important TF with roles in other tissues, beyond the lens, the implications of the findings in this manuscript have broader impact on the understanding of development. However, there are a few concerns that need to be addressed or made clear.

1. A rapid analysis of the 25 differentially expressed genes in *Gata3* CKO lenses (Table 1) in iSyTE demonstrates that all of these candidates are sharply up-regulated between stages E12.5 and E16.5 in normal lens development. This coincides with fiber cell differentiation and therefore offers support to the overall conclusions described in the manuscript that *Gata3* controls expression of genes associated with fiber cell differentiation in the lens. Consider including this analysis or describing it in the manuscript.
2. In Results, line 133, when they are first mentioned, list these 72 genes along with the nature of their change (reduced or upregulated at the fold-change, p-value, etc. in *Gata3* CKO lens) in a Supplemental Table.
3. It is not clear whether p27 and p57 were found to be reduced in the *Gata3* CKO lenses as per the RNA-seq analysis? Are these reduced on the RNA level by RT-qPCR (confirming the previous *in situ* hybridization data cited in the manuscript)?
4. In the Results, lines 181-182, increase in H3K27me3 signals was found in *Gata3* CKO lenses. It will be interesting to note if there was a stark difference in the numbers of up/down regulated genes in *Gata3* CKO lenses. Particularly, are there more up-regulated genes compared to down-regulated genes in the 72 DEGs? If not, what could be a potential explanation? Any other TF that is a potential downstream target of *Gata3* that may offer an explanation?
5. In Results, lines 186-187, with respect to the reduction of nuclear speckles in *Gata3* CKO lenses, is there evidence in the RNA-seq for mis-regulation of alternative splicing?

6. Consider describing the pattern of Gata3 expression in the lens (as reported previously) in the Introduction. This will add to reminding the readers of the importance of this Tf in the lens.

Minor points:

1. As a general note, the Gata3 lens-specific conditional knockout mice should be consistently referred as “Gata3 CKO” in the description of the data and discussion throughout the manuscript. On several occasions different nomenclatures are used (Gata3^{-/-}, Gata3 null; for example, see Figure 5 legend) when these are not meant to refer to Gata3 germline nulls but rather Gata3 CKOs.
2. In abstract, lines 5-6, “The rescued embryos showed defects in lens fiber cell differentiation”. This could be made more clear. Suggestion: “the rescue experiment could only partially rescue defects in Gata3^{-/-} mice, and did not rescue lens fiber defects. indicating that Gata3 played a role in the lens.
3. In Introduction, lines 45-46, mention some other key TFs in the lens, e.g. Sox1, c-Maf, Mafg/k. Also in Introduction, lines 51-52, consider using the term “expression patterns” rather than “expression domains”.
4. In Discussion, lines 194-195, suggest replacing “similar transcription factors” with a more general “lens transcription factors”. This is because similarity can be ambiguous, for example, Zeb2 exhibits higher expression in lens epithelium, while Prox1 has the opposite expression pattern. Therefore, deleting the word similar or replacing it with lens is suggested.
5. In Figure 1 legend, line 376: e14,5 should be E14.5.
6. In Supplemental Figure S5, provide more details in the legend on what do the numbers represent on the x-axis. Also mention that vertical lines between two or more points represent shared regulatory relationships.

Decision letter (RSOB-19-0220.R0)

21-Oct-2019

Dear Professor Cvekl,

We are pleased to inform you that your manuscript RSOB-19-0220 entitled "Transcriptomic analysis and novel insights into lens fiber cell differentiation regulated by Gata3" has been accepted by the Editor for publication in Open Biology. The reviewers have recommended publication, but also suggest some minor revisions to your manuscript. Therefore, we invite you to respond to the reviewers' comments and revise your manuscript.

Please submit the revised version of your manuscript within 14 days. If you do not think you will be able to meet this date please let us know immediately and we can extend this deadline for you.

To revise your manuscript, log into <https://mc.manuscriptcentral.com/rsob> and enter your Author Centre, where you will find your manuscript title listed under "Manuscripts with

Decisions." Under "Actions," click on "Create a Revision." Your manuscript number has been appended to denote a revision.

- 1) A text file of the manuscript (doc, txt, rtf or tex), including the references, tables (including captions) and figure captions. Please remove any tracked changes from the text before submission. PDF files are not an accepted format for the "Main Document".
- 2) A separate electronic file of each figure (tiff, EPS or print-quality PDF preferred). The format should be produced directly from original creation package, or original software format. Please note that PowerPoint files are not accepted.
- 3) Electronic supplementary material: this should be contained in a separate file from the main text and meet our ESM criteria (see <http://royalsocietypublishing.org/instructions-authors#question5>). All supplementary materials accompanying an accepted article will be treated as in their final form. They will be published alongside the paper on the journal website and posted on the online figshare repository. Files on figshare will be made available approximately one week before the accompanying article so that the supplementary material can be attributed a unique DOI.

Online supplementary material will also carry the title and description provided during submission, so please ensure these are accurate and informative. Note that the Royal Society will not edit or typeset supplementary material and it will be hosted as provided. Please ensure that the supplementary material includes the paper details (authors, title, journal name, article DOI). Your article DOI will be 10.1098/rsob.2016[last 4 digits of e.g. 10.1098/rsob.20160049].

- 4) A media summary: a short non-technical summary (up to 100 words) of the key findings/importance of your manuscript. Please try to write in simple English, avoid jargon, explain the importance of the topic, outline the main implications and describe why this topic is newsworthy.

Images

Data-Sharing

It is a condition of publication that data supporting your paper are made available. Data should be made available either in the electronic supplementary material or through an appropriate repository. Details of how to access data should be included in your paper. Please see <http://royalsocietypublishing.org/site/authors/policy.xhtml#question6> for more details.

Data accessibility section

Sincerely,
The Open Biology Team
mailto:openbiology@royalsociety.org

Board Member's Comments:

The paper was reviewed in depth and the reviewers have provided quite a lot of comments to improve it. Please try to address most of them and resubmit soon. Thanks for submitting your nice work.

Reviewer(s)' Comments to Author:

Referee: 1

Comments to the Author(s)

Martynova et al. explore the role of Gata3 in lens development using a tissue-specific knockout approach. They confirm published results that Gata3 is required for proper lens differentiation and that loss of Gata3 results in congenital cataract. They perform RNAseq and show that some putative Gata3 target genes are also altered in other models of congenital cataract mutants. They also confirm the published reductions in p27/cdkn1b and p57/cdkn1c in Gata3 knockout lenses, arguing that the effect may be direct. Finally they report changes in the chromatin organization of Gata3 knockout lenses.

The work is mostly well-done and an important confirmatory finding about Gata3 in the lens. The RNAseq data may add important mechanistic information about what functions of Gata3 are essential, although this is outside the body of this work. There are several major and minor concerns for this publication that should be addressed prior to the publication of the work:

- 1) The authors correctly utilize a number of controls to demonstrate a lens phenotype, including those containing the Cre transgene. Since expression of Cre itself can be toxic to lens cells (see Lam PT, et al. Hum Genomics. 2019 Feb 15;13(1):10), all phenotypic work in the lens should be done with a Cre/+ control, which is shown in Fig. 1 and 2. However other important experiments like RNAseq were performed in Cre- controls, despite the known transcriptional changes associated with Cre-expression in the lens.
- 2) The control lenses in Fig. 1 are too small and low-resolution to properly assess the nuclear structure. This is a concern for the Cre/+; Gata3fl/+ lens, which appear to show nuclear retention, although this may also be due to utilizing a peripheral section. This figure should be enhanced so that nuclear structure can be properly assessed in appropriate central sections from all genotypes.

- 3) The immunofluorescent data for p27/cdkn1b and p57/cdkn1c are suggestive of reductions in protein levels, but the overall evidence presented for these proteins is too low to be conclusive. Several points need to be addressed:
- i) If the authors wish to make the point that this is potentially transcriptional control, they need to determine why the mRNA levels for these genes are not decreased, at least as far as their RNAseq criteria go.
 - ii) Much of the reported regulation of p27/cdkn1b and p57/cdkn1c occurs post-translationally, which is not discussed in the paper. It would not be surprising if the observed reduction in protein is due to altered protein stability. This needs to be addressed, at minimum utilizing RNA expression versus protein levels. Preferably, experiments should be performed in lens cell culture models using Gata3 knockdown/CRISPR approaches and modulation of protein stability via proteasome inhibition.
 - iii) The overall presented immunofluorescence data is relatively low quality and low resolution. These images need to be enhanced, either using higher magnification, or better microscopy. Furthermore, some of the antibodies appear to have high background or non-specific staining in other tissues.
- 4) The common phenotypic results of Gata3, Prox1, and N-Myc knockout lenses and their partially overlapping transcriptomic changes can be due to functional relatedness, as proposed by the authors and their model in Figure 8, but can also be due to overlapping indirect developmental stunting. In this respect, the transcriptomic data is indicating a signature of an immature or blocked lens fiber cell. This can be partially assessed using a different kind of cataract that is caused by a mutation in a structural protein, like beta crystalline, rather than a transcriptional regulator. At minimum, the authors need to expand their discussion of potentially indirect findings.
- 5) It is unclear why chromatin IP analysis is being performed at P0, while the RNAseq data and immunofluorescence analysis for p27/cdkn1b and p57/cdkn1c was performed at E14.5. In the case of p27/cdkn1b and p57/cdkn1c, the protein staining should be performed at P0 as well as repeated at E14.5. If there are differences, the ChIP may need to be repeated at E14.5
- 6) The authors probe potential binding sites for Gata3 upstream of cdkn1b and cdkn1c. Is it clear that these are the relevant regulatory regions that control expression of these genes in the lens? Furthermore, what is the overall frequency of Gata3 sites within the genome, and are these sites likely to be by chance or because of specific enrichment. Because there are not transcriptional changes associated with these binding sites, the data is less compelling.
- 7) The data evaluating H3K27 tri-methylation and nuclear speckles is highly phenomenological and does not advance any specific hypothesis. Given that these may be markers of a developmentally blocked lens, it would be useful to evaluate in a different congenital cataract model to assess whether this is specific to Gata3 inactivation or not.
- 8) There are minor grammatical errors throughout the manuscript (e.g. page 10, line 184 and 187).

Referee: 2

Comments to the Author(s)

The manuscript by Martynova et al. focuses on characterization of lens defects on the phenotypic and molecular level in Gata3 (encoding a transcription factor) lens-specific conditional knockout mice. The motivation of this study was to understand Gata3 function in the lens. This was based on the expression pattern of Gata3 in the developing lens as well as a previous study describing a rescue experiment in Gata3 germline mice that still showed the lack of rescue of lens defects, in turn, suggesting its potential function in this tissue. The manuscript describes that Gata3 cKO lenses exhibit nuclear degradation defects and identifies 72 differentially expressed RNAs by RNA-seq. Furthermore, by comparative analysis with other RNA-seq data from defective lenses of various TF-deletion mice (e.g. Prox1, etc.), they identify 25 commonly regulated candidates. Additionally, by qChIP analyses, the authors uncover p27 and p57 as potential direct targets of

Gata3 in the lens through the identification of Gata3 occupancy in their potential regulatory regions. Overall, the manuscript is very well written and the data are clearly presented and provide a significant advance in the understanding of the role of Gata3 in autonomously controlling gene expression regulation in fiber cell differentiation during lens development. Further, these data identify several new targets of Gata3, with many of these targets representing new potential candidates that are excellent for further investigation in the lens. Because Gata3 is an important TF with roles in other tissues, beyond the lens, the implications of the findings in this manuscript have broader impact on the understanding of development. However, there are a few concerns that need to be addressed or made clear.

1. A rapid analysis of the 25 differentially expressed genes in Gata3 CKO lenses (Table 1) in iSyTE demonstrates that all of these candidates are sharply up-regulated between stages E12.5 and E16.5 in normal lens development. This coincides with fiber cell differentiation and therefore offers support to the overall conclusions described in the manuscript that Gata3 controls expression of genes associated with fiber cell differentiation in the lens. Consider including this analysis or describing it in the manuscript.

2. In Results, line 133, when they are first mentioned, list these 72 genes along with the nature of their change (reduced or upregulated at the fold-change, p-value, etc. in Gata3 cKO lens) in a Supplemental Table.

3. It is not clear whether p27 and p57 were found to be reduced in the Gata3 CKO lenses as per the RNA-seq analysis? Are these reduced on the RNA level by RT-qPCR (confirming the previous in situ hybridization data cited in the manuscript)?

4. In the Results, lines 181-182, increase in H3K27me3 signals was found in Gata3 CKO lenses. It will be interesting to note if there was a stark difference in the numbers of up/down regulated genes in Gata3 CKO lenses. Particularly, are there more up-regulated genes compared to down-regulated genes in the 72 DEGs? If not, what could be a potential explanation? Any other TF that is a potential downstream target of Gata3 that may offer an explanation?

5. In Results, lines 186-187, with respect to the reduction of nuclear speckles in Gata3 CKO lenses, is there evidence in the RNA-seq for mis-regulation of alternative splicing?

6. Consider describing the pattern of Gata3 expression in the lens (as reported previously) in the Introduction. This will add to reminding the readers of the importance of this Tf in the lens.

Minor points:

1. As a general note, the Gata3 lens-specific conditional knockout mice should be consistently referred as “Gata3 CKO” in the description of the data and discussion throughout the manuscript. On several occasions different nomenclatures are used (Gata3^{-/-}, Gata3 null; for example, see Figure 5 legend) when these are not meant to refer to Gata3 germline nulls but rather Gata3 CKOs.

2. In abstract, lines 5-6, “The rescued embryos showed defects in lens fiber cell differentiation”. This could be made more clear. Suggestion: “the rescue experiment could only partially rescue defects in Gata^{-/-} mice, and did not rescue lens fiber defects. indicating that Gata3 played a role in the lens.

3. In Introduction, lines 45-46, mention some other key TFs in the lens, e.g. Sox1, c-Maf, Mafg/k. Also in Introduction, lines 51-52, consider using the term “expression patterns” rather than “expression domains”.

4. In Discussion, lines 194-195, suggest replacing “similar transcription factors” with a more general “lens transcription factors”. This is because similarity can be ambiguous, for example, Zeb2 exhibits higher expression in lens epithelium, while Prox1 has the opposite expression pattern. Therefore, deleting the word similar or replacing it with lens is suggested.

5. In Figure 1 legend, line 376: e14,5 should be E14.5.

6. In Supplemental Figure S5, provide more details in the legend on what do the numbers represent on the x-axis. Also mention that vertical lines between two or more points represent shared regulatory relationships.

Author's Response to Decision Letter for (RSOB-19-0220.R0)

See Appendix A.

Decision letter (RSOB-19-0220.R1)

18-Nov-2019

Dear Professor Cvekl,

We are pleased to inform you that your manuscript entitled "Transcriptomic analysis and novel insights into lens fiber cell differentiation regulated by Gata3" has been accepted by the Editor for publication in Open Biology.

Article processing charge

Please note that the article processing charge is immediately payable. A separate email will be sent out shortly to confirm the charge due. The preferred payment method is by credit card; however, other payment options are available.

Sincerely,

The Open Biology Team
mailto: openbiology@royalsociety.org

Appendix A

Professor David Glover
Editor-in-Chief
Open Biology
University of Cambridge

November 19, 2019

RE: RSOB-19-0220

Dear Dr. Glover,

Thank you very much for your e-mail on October 21, 2019 notifying us about the provisional acceptance of our manuscript "Transcriptomic analysis and novel insights into lens fiber cell differentiation regulated by Gata3" for publication in *Open Biology*. We would like to thank all two reviewers for their helpful criticisms. We also acknowledge extended time given us to address all the points of criticism. We have changed our manuscript in response to all specific concerns and suggestions. The modified manuscript's text is shown in blue font.

Referee: 1

1) The authors correctly utilize a number of controls to demonstrate a lens phenotype, including those containing the Cre transgene. Since expression of Cre itself can be toxic to lens cells (see Lam PT, et al. *Hum Genomics*. 2019 Feb 15;13(1):10), all phenotypic work in the lens should be done with a Cre/+ control, which is shown in Fig. 1 and 2. However other important experiments like RNAseq were performed in Cre- controls, despite the known transcriptional changes associated with Cre-expression in the lens.

Response: Our system is based on the use of a different Pax6-cre line, not le-cre (see Lam et al. 2019, new reference included in the revision). These transgenes differ in the exact DNA sequences used, integration sites, and copy numbers. Nevertheless; they produce similar phenotypes following inactivation of Cbp and Ep300 genes as shown in our earlier study (Wolf et al. 2013). Whether or how Cre impairs lens precursor cells and early lens fibers remains unknown and we show all four genotypes (Figs. 1 and 2). Moreover; in the differentially expressed set of 72 genes in Gata3 CKO, only slight upregulation of Eda2r was commonly found with the le-cre hemizygous model (Lam et al. 2019). We modified the text (Discussion) to indicate these facts in response of referees' concern.

2) The control lenses in Fig. 1 are too small and low-resolution to properly assess the nuclear structure. This is a concern for the Cre/+; Gata3fl/+ lens, which appear to show nuclear retention, although this may also be due to utilizing a peripheral section. This figure should be enhanced so that nuclear structure can be properly assessed in appropriate central sections from all genotypes.

Response: We agree on this point and changed the design of the original Fig. 1 and now broke it to Figs. 1 and 2. In new Fig. 2, all panels were enlarged to provide additional details. Furthermore, retention of nuclei is corroborated by reduced expression of Dnase2b and earlier data by Maeda et al. 2009.

3) The immunofluorescent data for p27/cdkn1b and p57/cdkn1c are suggestive of reductions in protein levels, but the overall evidence presented for these proteins is too low to be conclusive. Several points need to be addressed:

i) If the authors wish to make the point that this is potentially transcriptional control, they need to determine why the mRNA levels for these genes are not decreased, at least as far as their RNAseq criteria go.

ii) Much of the reported regulation of p27/cdkn1b and p57/cdkn1c occurs post-translationally, which is not discussed in the paper. It would not be surprising if the observed reduction in protein is due to altered protein stability. This needs to be addressed, at minimum utilizing RNA expression versus protein levels. Preferably, experiments should be performed in lens cell culture models using Gata3 knockdown/CRISPR approaches and modulation of protein stability via proteasome inhibition.

iii) The overall presented immunofluorescence data is relatively low quality and low resolution. These images need to be enhanced, either using higher magnification, or better microscopy. Furthermore, some of the antibodies appear to have high background or non-specific staining in other tissues.

Response: We thank the reviewer for raising these points that require further clarifications. i) In Discussion, we comment that the bulk RNA-seq data at E14.5 stage do not show significant changes at the mRNA levels; however, these transcriptional changes previously reported by Maeda et al. 2009 occur in a larger number of cells 48 hours later as shown by their qRT-PCR and immunofluorescence data at E16.5. At E14.5, it appears that only a small portion of lens fibers in Gata3 CKO are affected and bulk RNA-seq may not identify these changes. ii) The posttranslational regulatory mechanisms (new references Larocgue et al. 2005; Lin et al. 2019) are now included in the revised manuscript. For the lens model, a new reference was added regarding Celf1 and stability of p27 mRNAs (Siddam, Lachke et al. 2018). The experiments proposed by the reviewer to further probe the molecular mechanisms of p27/Cdkn1b and p57/Cdkn1c are in our opinion beyond the current scope of this report. iii) The images were improved (size and contrast).

4) The common phenotypic results of Gata3, Prox1, and N-Myc knockout lenses and their partially overlapping transcriptomic changes can be due to functional relatedness, as proposed by the authors and their model in Figure 8, but can also be due to overlapping indirect developmental stunting. In this respect, the transcriptomic data is indicating a signature of an immature or blocked lens fiber cell. This can be partially assessed using a different kind of cataract that is caused by a mutation in a structural protein, like beta crystallin, rather than a transcriptional regulator. At minimum, the authors need to expand their discussion of potentially indirect findings.

Response: We agree on this point and revised the text accordingly (Results and Discussion).

5) It is unclear why chromatin IP analysis is being performed at P0, while the RNAseq data and immunofluorescence analysis for p27/cdkn1b and p57/cdkn1c was performed at E14.5. In the case of p27/cdkn1b and p57/cdkn1c, the protein staining should be performed at P0 as well as repeated at E14.5. If there are differences, the ChIP may need to be repeated at E14.5.

Response: The qChIP experiments were conducted with newborn lenses as the single experiment needs as many as 40 lenses. For the entire experiment, we initially collected 400 lenses for 10 individual experiments. Use of E14.5 microdissected lenses to obtain sufficient amounts of the chromatin would require over 1,000 embryonic lenses per single assay which is not technically feasible.

6) The authors probe potential binding sites for Gata3 upstream of cdkn1b and cdkn1c. Is it clear that these are the relevant regulatory regions that control expression of these genes in the lens? Furthermore, what is the overall frequency of Gata3 sites within the genome, and are these sites likely to be by chance or because of specific enrichment. Because there are not transcriptional changes associated with these binding sites, the data is less compelling.

Response: It is not known in present how p27/Cdkn1b and p57/Cdkn1c genes are regulated through individual *cis*-sites in the lens. The only available experiment on Hey1/Herp2 with p57 upstream region is quoted in the text and illustrated in Fig. 9. Our data propose a testable model for future studies.

7) The data evaluating H3K27 tri-methylation and nuclear speckles is highly phenomenological and does not advance any specific hypothesis. Given that these may be markers of a developmentally blocked lens, it would be useful to evaluate in a different congenital cataract model to assess whether this is specific to Gata3 inactivation or not.

Response: We keep this Fig. 8 (new numbering) to illustrate a number of novel ways how to analyze mouse lens mutants that show the denucleation defects. As lens fiber cell denucleation remains poorly understood, we prefer to retain this section within the revised manuscript.

8) There are minor grammatical errors throughout the manuscript (e.g. page 10, line 184 and 187).

Response: Corrected.

Referee 2:

1. A rapid analysis of the 25 differentially expressed genes in Gata3 CKO lenses (Table 1) in iSyTE demonstrates that all of these candidates are sharply up-regulated between stages E12.5 and E16.5 in normal lens development. This coincides with fiber cell differentiation and therefore offers support to the overall conclusions described in the manuscript that Gata3 controls expression of genes associated with fiber cell differentiation in the lens. Consider including this analysis or describing it in the manuscript.

Response: We thank the reviewer for this suggestion and edited the manuscript accordingly (Results).

2. In Results, line 133, when they are first mentioned, list these 72 genes along with the nature of their change (reduced or upregulated at the fold-change, p-value, etc. in Gata3 cKO lens) in a Supplemental Table.

Response: A new Supplemental Table S3 has been added.

3. It is not clear whether p27 and p57 were found to be reduced in the Gata3 CKO lenses as per the RNA-seq analysis? Are these reduced on the RNA level by RT-qPCR (confirming the previous *in situ* hybridization data cited in the manuscript)?

Response: The bulk RNA-seq data at E14.5 did not reveal such transcriptomic changes; however, the Maeda et al. 2009 study analyzed their expression at E16.5 (see also response to Referee 1). The Results were edited to clearly state these comparative findings.

4. In the Results, lines 181-182, increase in H3K27me3 signals was found in Gata3 CKO lenses. It will be interesting to note if there was a stark difference in the numbers of up/down regulated genes in Gata3 CKO lenses. Particularly, are there more up-regulated genes compared to down-regulated genes in the 72 DEGs? If not, what could be a potential explanation? Any other TF that is a potential downstream target of Gata3 that may offer an explanation?

Response: This is the first observation that H3K27me3 staining change in any mutated lens fiber cell nuclei undergoing denucleation. Previously we showed that lens fiber cell nuclei transfer histone proteins into the cytoplasm during their condensation (Limi et al. 2018) as previously shown for mammalian erythrocytes. We agree with the reviewer that the most likely consequence of this pattern include chromatin condensation followed by transcriptional

repression (detected for 29 from 72 genes); nevertheless, this initial observation requires deeper studies that are beyond the scope of the present report. The Discussion was edited to introduce these points.

5. In Results, lines 186-187, with respect to the reduction of nuclear speckles in Gata3 CKO lenses, is there evidence in the RNA-seq for mis-regulation of alternative splicing?

Response: This is a valuable point and the data can be further scrutinized as shown earlier in the lens (Srivastava et al. 2017). The Discussion was edited accordingly.

6. Consider describing the pattern of Gata3 expression in the lens (as reported previously) in the Introduction. This will add to reminding the readers of the importance of this TF in the lens.

Response: We thank the reviewer for this point and the Introduction was extended to improve this point.

Minor points:

1. As a general note, the Gata3 lens-specific conditional knockout mice should be consistently referred as “Gata3 CKO” in the description of the data and discussion throughout the manuscript. On several occasions different nomenclatures are used (Gata3^{-/-}, Gata3 null; for example, see Figure 5 legend) when these are not meant to refer to Gata3 germline nulls but rather Gata3 CKOs.

Response: The entire manuscript was checked and Gata3 CKO is used.

2. In abstract, lines 5-6, “The rescued embryos showed defects in lens fiber cell differentiation”. This could be made more clear. Suggestion: “the rescue experiment could only partially rescue defects in Gata3^{-/-} mice, and did not rescue lens fiber defects. indicating that Gata3 played a role in the lens.

Response: Revised Abstract as suggested.

3. In Introduction, lines 45-46, mention some other key TFs in the lens, e.g. Sox1, c-Maf, Mafg/k. Also in Introduction, lines 51-52, consider using the term “expression patterns” rather than “expression domains”.

Response: Revised as suggested.

4. In Discussion, lines 194-195, suggest replacing “similar transcription factors” with a more general “lens transcription factors”. This is because similarity can be ambiguous, for example, Zeb2 exhibits higher expression in lens epithelium, while Prox1 has the opposite expression pattern. Therefore, deleting the word similar or replacing it with lens is suggested.

Response: Corrected as suggested.

5. In Figure 1 legend, line 376: e14,5 should be E14.5.

Response: Typo corrected.

6. In Supplemental Figure S5, provide more details in the legend on what do the numbers represent on the x-axis. Also mention that vertical lines between two or more points represent shared regulatory relationships.

Response: Edited as suggested.

If any questions or problems should arise concerning this manuscript, please do not hesitate to contact me.

Sincerely,

Ales Cvekl, Ph.D.